

# Effects of Arctic sea-ice concentration on surface radiative fluxes in four atmospheric reanalyses

Tereza Uhlíková[1,2], Timo Vihma[2], Alexey Yu Karpechko[2], and Petteri Uotila[1]

[1]Institute for Atmospheric and Earth System Research, Faculty of Science, University of Helsinki, 00014, Helsinki, Finland
[2]Finnish Meteorological Institute, Helsinki, Finland

**Correspondence:** Tereza Uhlíková (tereza.uhlikova@helsinki.fi)

**Abstract.** Spatio-temporal variations and climatological trends in the sea-ice concentration (SIC) are highly important for the energy budget of the lower atmosphere and the upper ocean in the Arctic. To better understand the local, regional, and global impacts of the recent rapid sea-ice decline, one of the key issues is to quantify the interactions of SIC and the surface radiative fluxes. We analyse these effects utilising four global atmospheric reanalyses, ERA5, JRA-55, MERRA-2, and NCEP/CFSR and
evaluate the uncertainties arising from inter-reanalysis differences in the sensitivity of the surface radiative fluxes to SIC. Using daily data over the period 1980–2021, the linear orthogonal-distance regression indicates similar sensitivity of surface upward longwave radiation to SIC in all reanalyses with the greatest sensitivity in the cold season November–April (over 150 W m$^{-2}$ per -0.1 change in SIC) and up to 80 W m$^{-2}$ per -0.1 change in SIC in May–October. We find that the effect of SIC on both surface upward longwave and shortwave radiation has mostly weakened in all seasons between the study periods of 1980–2000
and 2001–2021. The decrease in the sensitivity of upward longwave radiation to SIC can be attributed to the increasing surface temperature of sea ice, which dominated in the inner ice pack, and to the sea-ice decline, which dominated in the marginal ice zone. Approximately 80 % of the decadal decrease in upward shortwave radiation in May–July was caused by a decrease in surface albedo, controlled by SIC decrease, and the rest was caused by a decrease in downward shortwave radiation due to increase in cloudiness, mostly close to sea ice margins.

## 1  Introduction

Sea ice in the Arctic Ocean both affects and is affected by thermal longwave radiation and solar shortwave radiation. The former dominates the surface net radiation over most of the year and triggers the spring onset of snowmelt on top of sea ice (Mortin et al., 2016), whereas the latter is the key driver of summertime surface melt of snow and ice (Perovich et al., 2007). In winter over the Arctic Ocean, the snow surface temperature occasionally drops below -40 °C, which strongly reduces the
emitted longwave radiation (Persson et al., 2002). Simultaneously, open leads with a surface temperature close to -1.8 °C emit almost double the amount of longwave radiation, and refrozen leads have intermediate values for surface temperature and longwave radiation emission.

In summer, the surface conditions are close to isothermal, and the longwave radiation emitted is much less sensitive to the presence of sea ice, whereas the effects of sea ice and snow on reflected solar radiation are strong. New dry snow has a





surface albedo of approximately 0.85, and even melting ice has a surface albedo of approximately 0.4 (Light et al., 2022), which is much higher than that of the open sea (less than 0.1). Hence, during spring and summer, the strong reflection from the snow or ice surface strongly reduces the surface net shortwave radiation. Throughout the year, both open water and sea ice surfaces generally emit more longwave radiation than they receive from clouds and the atmosphere (Persson, 2012). This is due to the high emissivity of snow and ice, 0.97-0.98 (Liang et al., 2014), which far exceeds the typical emissivity of the Arctic

atmosphere, even under cloudy conditions (Garrett and Zhao, 2006). An exception occurs in the presence of thick water clouds in summer, which emit almost like a black body and have base temperatures close to or even higher than that of the snow/ice surface.

The above-mentioned findings are based on data from rare field campaigns in the Arctic sea ice zone. To understand the processes on a regional scale as well as their seasonal, inter-annual, and decadal variations and past trends, atmospheric and

ocean reanalyses, as well as satellite remote sensing products, must be applied. Comparison of different reanalyses against each other and observations is vital to evaluate their uncertainty. Reanalysis products for surface radiative fluxes over sea ice have been compared and evaluated in several studies (Walsh et al. (2009); Graham et al. (2019); Jonassen et al. (2019); Yeo et al. (2022)). The ERA5 (Hersbach et al., 2020) and NCEP/CFSR (Saha et al. (2010); Saha et al. (2014)) reanalyses generally perform better than others (Jonassen et al. (2019); Di Biagio et al. (2021)), but challenges remain, especially for clouds and

downward longwave radiation in winter (Graham et al., 2019). Additionally, reanalysis products for sea-ice concentration (SIC) have been compared (Graham et al., 2019). However, we are not aware of any study addressing inter-reanalysis differences in the relationship between SIC and radiative surface fluxes. This is a key question, as SIC plays a crucial role in the radiative surface fluxes and the surface energy balance over the Arctic Ocean.

Relevant research questions include the spatial patterns of the relationships between SIC and radiative surface fluxes over

the Arctic Ocean, and the seasonal evolution of these relationships during the spring and autumn transitions. Considering the threshold value of SIC for sea ice to dominate the sign of the regional surface fluxes, it is known that for turbulent surface fluxes in winter, the threshold typically exceeds 0.9 (Vihma (1995); Andreas et al. (2010)), but for radiative fluxes, the threshold has not received as much attention. Regarding climatological trends, according to satellite passive-microwave data from 1979—2021, the average yearly sea-ice extent in the Arctic has declined by more than 50 000 km$^2$ per year (Parkinson, 2022).

To understand at the process level how the major sea ice decline has affected the ocean and atmosphere locally, regionally, and globally, the necessary first step is to quantify the effects of SIC on the surface energy balance of the Arctic Ocean. Furthermore, the range of uncertainty in these effects and their changes over recent decades deserves attention.

To meet the above-mentioned challenges, we analyze the effects of SIC on surface upward shortwave and longwave radiation and clouds based on products of four atmospheric reanalyses. This is a follow-up study to Uhlíková et al. (2024), in which

we addressed the effects of SIC on the turbulent surface fluxes of sensible and latent heat over the Arctic Ocean.



## 2 Material and Methods

To investigate the relationship between SIC and radiative surface fluxes, we utilised data from four atmospheric reanalyses. Because this paper is a companion paper to Uhlíková et al. (2024) (hereafter referred to as 'the companion paper'), we use data from (1) the same reanalyses (ERA5 (Hersbach et al., 2023), JRA-55 (JMA, 2013), MERRA-2 (GMAO (2015a);
GMAO (2015b); GMAO (2015c)), NCEP/CFSR (Saha et al. (2010), Saha et al. (2011))), (2) the same study periods (1980–2000 and 2001–2021), (3) the same seasons (November–December–January, February–March–April, May–June–July, August–September–October), and the same temporal resolution (daily means of data), to make the two studies comparable. The term 'NCEP/CFSR' refers to data from both NCEP Climate Forecast System Reanalysis (CFSR; covering the period 1980–2010, spatial resolution 0.312° lat ×0.313° lon) and NCEP Climate Forecast System Version 2 (CFSv2; covering the period 2011–
2021, spatial resolution 0.204°×0.205°). We unified the spatial resolution for the whole 'NCEP/CFSR' data set to 0.4°×0.4° using bilinear interpolation. Besides this adjustment, we worked with the original horizontal spatial resolution of the remaining reanalyses: 0.25°×0.25° (ERA5), 0.561°×0.563° (JRA-55), and 0.5°×0.625° (MERRA-2).

From each reanalysis, we have used the following variables: sea-ice concentration (SIC), surface upward longwave radiation (ULW), surface temperature ($T_s$), surface upward shortwave radiation (USW), surface downward shortwave radiation
(DSW), and cloud water (vertically integrated cloud liquid water + cloud ice; hereafter referred to 'cloud condensate content', CCC). We chose CCC as a metric for cloud conditions, as it provides better available estimate of cloud radiative properties compared to total cloud cover (Senkova et al., 2007).

Using these data, we studied bilateral relationships between SIC and surface upward radiative fluxes (ULW, USW) utilizing linear bilateral orthogonal-distance-regression model (ODR; Boggs et al. (1988)). Because all variables in reanalyses include
uncertainties, ODR model is more optimal for this data than ordinary-least-squares-regression model (OLSR), which assumes no uncertainty in the independent variable (in our case SIC). Additionally, we performed a comparison study of bilateral ODR and OLSR outputs using data from the above-mention reanalyses and noted, that while the coefficients of determination ($R^2$) were 'nearly identical' (at least to five decimal points identical) for both methods, the values of slopes of the regression line varied considerably. Based on these findings, we additionally decided to utilize OLSR analyses when only studying $R^2$, as
this regression method requires less computing resources to perform. We used linear model for both ODR and OLSR as we evaluated it as the most applicable for our purposes primarily following from the finding that typically the first order i.e. linear term dominates over higher order ones when describing the relationship between two variables with the Taylor series.

The statistical-significance testing of the results was performed using Student's t-test (95 % confidence interval) with adjusted degrees of freedom ($DF_{adj}$) according to Eq. (31) from Bretherton et al. (1999) to account for autocorrelation of the time
series:

$$DF_{adj} = T \frac{1 - R_1 R_2}{1 + R_1 R_2} \tag{1}$$

where T stands for number of days in one sample (in our case days in seasons in the periods of 1980–2000 or 2001–2021) and $R_1$ respectively $R_2$ for correlation coefficient for lag 1 auto-correlation of surface radiative flux (ULW, USW) and its explanatory variable (SIC). To test the field statistical significance of the coefficients of determination (OLSR) and differences





**Table 1.** Parameterisation of surface albedo and representation of the sea ice in reanalyses.

|  | ERA5 | JRA-55 | MERRA-2 | NCEP/CFSR |
|---|---|---|---|---|
| **Sea-ice albedo** | Prescribed seasonal cycle, based on Ebert and Curry (1993) | Parameterised, function of hourly $\theta_s$[a] and $T_s$[b] | Prescribed seasonal cycle, based on Duynkerke and de Roode (2001) | Parameterised |
| **Sea-ice thickness** | 1.5 m, fixed | 2 m, fixed | n/a[c] | Modelled (coupled) |
| **Snow on sea ice** | None | None | None | Modelled (coupled) |
| **Sea-ice concentration** | Fractional, external data set (OSI SAF[e] (409a) 1979/Aug 2007, OSI SAF[e] oper Sep 2007-) | Binary[d], external data set (COBE-SST[f]) | Fractional, external data set (OISST[g] 1982/Mar 2006, OSTIA[h] Apr 2006-) | Fractional, modelled (coupled) |

[a] Solar zenith angle. [b] Surface temperature. [c] A 7-cm ice layer for computing a prognostic ice surface temperature, which is then relaxed towards 273.15 K as a representation of the upward oceanic heat flux; n/a: not applicable. [d] SIC > 0.55 = 1, SIC ≤ 0.55 = 0. [e] Ocean and Sea Ice Satellite Application Facility. [f] Centennial In Situ Observation-based Estimates of the Variability of Sea Surface Temperatures and Marine Meteorological Variables. [g] Optimum Interpolation Sea Surface Temperature. [h] Operational Sea Surface Temperature and Ice Analysis.

in mean decadal seasonal values between the two study periods, we have used p-value $< 0.05$ adjusted by $\alpha_{\mathrm{FDR}} = 0.10$ (false discovery rate, according to Wilks (2016)) to reject the null-hypothesis that the time series are independent.

As we concluded in the companion paper, the largest differences in the effects of Arctic SIC on surface turbulent fluxes in reanalyses come from the representation of the sea ice, which is modelled in NCEP/CFSR and prescribed in ERA5, JRA-55, and MERRA-2. In Table 1, we reiterate the most important differences in representation of the sea ice in reanalyses and

95 furthermore present differences in parameterisation of the sea-ice albedo.





## 3 Results

### 3.1 Effects of sea-ice concentration on the surface upward longwave radiative flux

Utilizing linear bilateral ODR analysis, we assessed the effects of SIC on ULW. These two variables were negatively correlated in all seasons and both study periods (Figs. 1 and S1, S3, S4), meaning less SIC–more ULW or more SIC–less ULW. The sign of the correlation was in agreement with the theoretical expectations as the open ocean surface in the Arctic is usually warmer than the sea-ice surface (and much warmer in the cold season, November–April), and accordingly emits more longwave radiation. As depicted in the above-mentioned Figures, the sensitivity of ULW to SIC (slope of the regression line) did not vary considerably among reanalyses, with the highest values over 150 W m$^{-2}$ ULW per -0.1 change in SIC in November–April in the Central Arctic (north of 81.5° N). The dark grey areas indicate a failure of the linear bilateral ODR model to converge. For JRA-55 (Fig. 1b, f, j), this was caused by the binary representation of SIC in this reanalysis, which assigns value 1 to SIC > 0.55, and value 0 to SIC ≤ 0.55. Then, because the SIC in these dark grey areas was never less than 0.55 during the 21-year periods, every grid cell was assigned a value of 1. Hence, no dependence with ULW or any other variable could be found. In other reanalyses, the ODR model failure also occurred either because of very low variability in SIC or due to high uncertainty in the slope of regression between the two variables as shown in Figs. S1 and S2.

The sensitivity of ULW to SIC mostly decreased in all seasons between 1980–2000 and 2001–2021 (shades of red in panels i–l in Figs. 1 and S1, S3, S4), but strengthened in the Central Arctic (shades of blue panels i–l in Figs. 1 and S1, S3, S4). To explain these changes, in Fig. 2, we show the daily SIC and ULW in grid cells from ERA5, MERRA-2, and NCEP/CFSR data where the sensitivity changed considerably between 1980–2000 and 2001–2021 in November–December–January. While in Point 1 (see Fig. 1) from the border of Chukchi and East Siberian seas, the slope of the regression line became less steep in 2001–2021 compared to 1980–2000, in Point 2 from the Central Arctic, the slope became steeper in the second (more recent) study period.

As shown in Uhlíková et al. (2024, Fig. 5), the surface temperature of the Arctic sea ice generally increased between the two study periods, therefore, the difference between the surface temperature of the ice ($T_{ice}$) and the ocean decreased causing lower sensitivity of ULW to SIC in the majority of the Arctic in all seasons in the second study period. Also in this study, we show in Fig. 2: Point 1, that ULW (and therefore the surface temperature) is generally higher in 2001–2021 (lower panels) than 1980–2000 (upper panels) in days with SIC = 1. Another cause of decreasing sensitivity of ULW to SIC is the fact that in areas where the SIC declined or disappeared completely between the two study periods, there is naturally smaller or no effect of SIC on ULW in the second study period. ULW is also generally not so sensitive to SIC in regions where SIC is low, because in such regions, $T_{ice}$ is typically higher, closer to sea-surface temperature. This is illustrated in the lower panels of Fig. 2: Point 1, where all the values of ULW in the grid cells with SIC lower than approximately 0.5 fluctuate close to 300 W m$^{-2}$.

The increased sensitivity of ULW to SIC in smaller areas in the Central Arctic may be due to increased SIC in reanalyses in these areas in 2001–2021 compared to 1980–2000. As shown in Fig. 2: Point 2, there are indeed both higher SIC as well as steeper slopes of the regression lines in the second study period (lower panels) than in the first one (upper panels). We discuss the possible mechanisms of the increased SIC in Section 4.1.



**Figure 1.** Change in upward longwave radiative flux (W m$^{-2}$) per change of 0.1 in sea-ice concentration (slope of regression line) in the marine Arctic in November–December–January in four reanalyses (columns), based on the linear orthogonal-distance-regression (ODR) model. Dark grey indicates areas where the ODR model did not converge; in panels **(i)–(l)**, dark grey shows these areas in 1980–2000 and/or 2001–2021. Only grid cells with a mean of SIC > 0.5 were considered, and only the slopes whose 95 % confidence intervals do not overlap zero are shown (others masked in white). Points 1 and 2 (in black) from panels **(i), (k), (l)** are further analysed in Figure 2.





**Figure 2.** Daily sea-ice concentration (SIC) and upward longwave radiative flux (ULW) in selected grid cells, indicated in Figure 1 in panels **(i), (k), (l)**, where the sensitivity of ULW to SIC between 1980—2000 and 2001-–2021 decreased (Point **1**, grid cell nearest to 73° N, 180° W) and increased (Point **2**, grid cell nearest to 83° N, 0° W). ERA5, MERRA-2, and NCEP/CFSR data, days in November–December–January (1932 days). Black solid lines depict (a part of) the regression line and illustrate their slope.





To further explore the effect of the surface type in the marine Arctic on ULW, we investigated whether the main driver
of ULW is the SIC or the surface temperature of sea ice ($T_{ice}$). To answer this question, we compared $R^2$ (coefficient of deter-
mination) using SIC and $T_{ice}$ as explanatory variables for ULW. To calculate $T_{ice}$ from the grid-averaged surface temperature
($T_s$), we utilized the following equation:

$$T_{ice} = \frac{T_s - (1 - SIC)T_{ocean}}{SIC} \tag{2}$$

where we assumed temperature of the ocean ($T_{ocean}$) at -1.8 °C (271.35 K). Naturally, this assumption cannot be applied in
all grid cells, where SIC is present, especially in the Atlantic sector (Greenland and Barents seas), and in warm season (May–
October) in the majority of adjacent seas outside the Central Arctic. Therefore, we focused on the cold season (November–
April) in these analyses. We utilized data from only ERA5, MERRA-2, and NCEP/CFSR because JRA-55 comes with binary
representation of SIC, hence, Eq. (2) is not applicable for this data set. As shown in Figs. 3 and S5, in November–April, $T_{ice}$
explains over 90 % of the variance of ULW (vULW) in areas, where SIC is very high, whereas SIC explains only around 30 %
of vULW in these areas. However, in the marginal ice zone, the coefficient of determination was higher for SIC (around 60 %)
compared to $T_{ice}$ ($< 30$ %). These results were quantitatively very similar in both study periods.







**Figure 3.** Proportion of variance in the upward longwave radiation (ULW) explained by sea-ice concentration (SIC) and surface temperature of the ice ($T_{ice}$) in November–December–January, 1980–2000 and 2001–2021 (columns), as represented in three reanalyses (rows), based on linear ordinary-least-square regression model (coefficient of determination, $R^2$) using daily means of data from ERA5, MERRA-2, and NCEP/CFSR. Only grid cells with a mean of SIC > 0.5 were considered and only statistically significant results at the 5 % level of significance are shown (insignificant masked in white).



## 3.2 Effects of sea-ice concentration on surface upward shortwave radiative flux

The sea-ice surface (bare or snow-covered) has much higher shortwave reflectivity than the open sea. Hence, as expected, we
found a positive correlation between SIC and USW meaning more SIC–more USW or less SIC–less USW in all seasons with
solar radiation present in the Arctic (Figs. 4, S6, S8). USW was the most sensitive to SIC in May–June–July in the Central
Arctic – over 100 W m$^{-2}$ USW per 0.1 change in SIC. The ODR model did not converge in large ares of the marine Arctic in
February–March–April and August–September–October due to lack of variability in both incoming solar radiation, which was
mostly very low during these months, and in SIC, which was very high. This is illustrated for representative grid cells in Figs.
S6 and S7.

The effect of SIC on USW weakened between 1980–2000 and 2001–2021 in nearly all of the Arctic (shades of blue in
panels i–l in Figs. 4 and S6). As discussed in the previous subsection, the sea-ice decline in adjacent Arctic seas naturally
contributes to decreased effect of SIC on ULW; the same applies also to SIC effect on USW. However, because USW is a
result of both, the downward shortwave radiation (DSW) and the reflectivity of the surface (surface albedo), the decrease in
USW sensitivity to SIC between the study periods could have been additionally caused by changes in either or both of its
above-mentioned drivers. To address this issue, we created Figs. 5, 6, S9–S12, which show changes in seasonal means of
shortwave radiative fluxes between the periods ($\Delta$DSW, $\Delta$USW), $\Delta$USW explained by change in DSW ($\Delta$USW$_{DSW}$), and
$\Delta$USW explained by change in surface albedo (b, $\Delta$USW$_b$). The above-mentioned variables were calculated for each grid cell
using daily data according to the following equations:

$$\Delta DSW = DSW_{2001-2021 \text{ mean}} - DSW_{1980-2000 \text{ mean}} \tag{3}$$

$$\Delta USW = USW_{2001-2021 \text{ mean}} - USW_{1980-2000 \text{ mean}} \tag{4}$$

$$b = \frac{USW_{1980-2000 \text{ mean}}}{DSW_{1980-2000 \text{ mean}}} \tag{5}$$

$$\Delta USW_{DSW} = b \times \Delta DSW \tag{6}$$

$$\Delta USW_b = \Delta USW - \Delta USW_{DSW} \tag{7}$$

For May–June–July, Fig. 5a–d shows that reanalyses agreed on the strongest decline (around -15 W m$^{-2}$) in the mean
DSW between 1980–2000 and 2001–2021 in northern Barents Sea between Svalbard and Novaja Zemlya and some smaller





**Figure 4.** Change in upward shortwave radiative flux (W m$^{-2}$) per 0.1 change in sea-ice concentration (slope of regression line) in four reanalyses (columns), marine Arctic, November–December–January, based on the linear orthogonal-distance-regression (ODR) model. Dark grey indicates areas where the ODR model did not converge; in panels **(i)–(l)**, dark grey shows these areas in 1980–2000 and/or 2001–2021. Only grid cells with a mean of SIC > 0.5 were considered, and only the slopes whose 95 % confidence intervals do not overlap zero are shown (others masked in white).

degree of decline in this variable in other adjacent Arctic seas. All reanalyses also agreed on an increase around 10 W m$^{-2}$ in



the mean DSW between 1980–2000 and 2001–2021, north of Greenland and Canadian archipelago. However, the areal extent of this increase varied considerably between the data sets, with NCEP/CFSR showing the largest one, followed by MERRA-2.

According to Fig. 6 row i, the areas of increased DSW correspond with those where CCC (vertically integrated cloud water + ice) diminished between the two study periods. Vice versa, the area of strongest decadal seasonal reduction of DSW in northern Barents Sea between Svalbard and Novaja Zemlya can be connected with the one where CCC increased.

Mean USW between 1980–2000 and 2001–2021 (Fig. 5e–h) declined in most of adjacent Arctic seas by more than -25 W m$^{-2}$ in all reanalyses. In agreement with theoretical expectations, most of the decadal seasonal reduction in USW outside the

175 Central Arctic (around 80 %) was attributed to decrease in surface albedo (shades of blue in Figs. 5m–p and 6 row ii) which to a large part coincided with SIC decline (shades of blue in Fig. 6 row iii). However, also reduction of DSW (around -5 W m$^{-2}$) played a role (Fig. 5i–l). ERA5 and NCEP/CFSR indicated an increase in mean USW (around +10 W m$^{-2}$) in 2001–2021 in the Central Arctic, north of Greenland and Canadian archipelago (shades of red in Fig. 5e, h) which spread about equally between an increase in albedo and DSW in this area (shades of red in Fig. 5i, l, m, p). As the sea-ice albedo in ERA5 has a

180 prescribed seasonal cycle, the same for both study periods, the decadal seasonal increase in albedo can only be explained by increased SIC in the area (as shown in Fig. 6 row iii). In NCEP/CFSR however, albedo is parameterized, therefore the increase in this variable indicates decadal seasonal changes in surface albedo in this data set (Fig. 6 row ii). At the same time, we also detected decadal seasonal growth in SIC in the Central Arctic in NCEP/CFSR (Fig. 6 row iii).

In February–March–April, we found very little statistically significant decadal differences in DSW, however, reanalyses

generally agreed that there was an increase in CCC over the Barents Sea, between Svalbard and Novaja Zemlya, and decline along the east coast of Greenland (Fig. S10 row i). We found mostly decadal reduction in USW (around -15 W m$^{-2}$) in the marginal ice zone (shades of blue, Fig. S9e–h). This reduction, similarly to May–June–July, was mostly attributed to decline in surface albedo (Fig. S9m–p), but partly also to reduction in DSW (Fig. S9i–l).

In August–September–October, we noted decadal reduction in mean DSW around -10 W m$^{-2}$ in adjacent Arctic seas. All

190 reanalyses also agreed on decadal reduction in the mean USW though disagreed on the magnitude over the Beaufort, Chukchi, East Siberian, and Laptev seas. In these areas, the decrease in USW ranged between around -20 W m$^{-2}$ in JRA-55 and around -10 W m$^{-2}$ in MERRA-2 (Fig. S11). As in the two previously-mentioned seasons, more of the mean USW reduction between 1980–2000 and 2001–2021 was attributed to decline in surface albedo than decline in DSW. Regarding decadal changes in mean CCC, we found a strong increase across the Arctic, though reanalyses showed a large scatter on the magnitude and

195 spatial pattern of this change (Fig. S12 row i).







**Figure 5.** Changes in decadal means (calculated from daily means) between 1980–2000 and 2001–2021, May–June–July. Panels **(a)–(h)** show changes in surface downward and upward shortwave radiative fluxes (ΔDSW, ΔUSW), panels **(i)–(l)** show changes in USW explained by changes in DSW (Δ USW$_{DSW}$), and panels **(m)–(p)** changes in USW explained by changes in albedo (Δ USW$_b$). Only statistically significant results at the 5 % level of significance are shown (insignificant masked in white); statistically significant grid cells for ΔUSW, Δ USW$_{DSW}$, and Δ USW$_{DSW}$ are identical. Values within an interval (-0.1,0.1) W m$^{-2}$ are also masked in white.

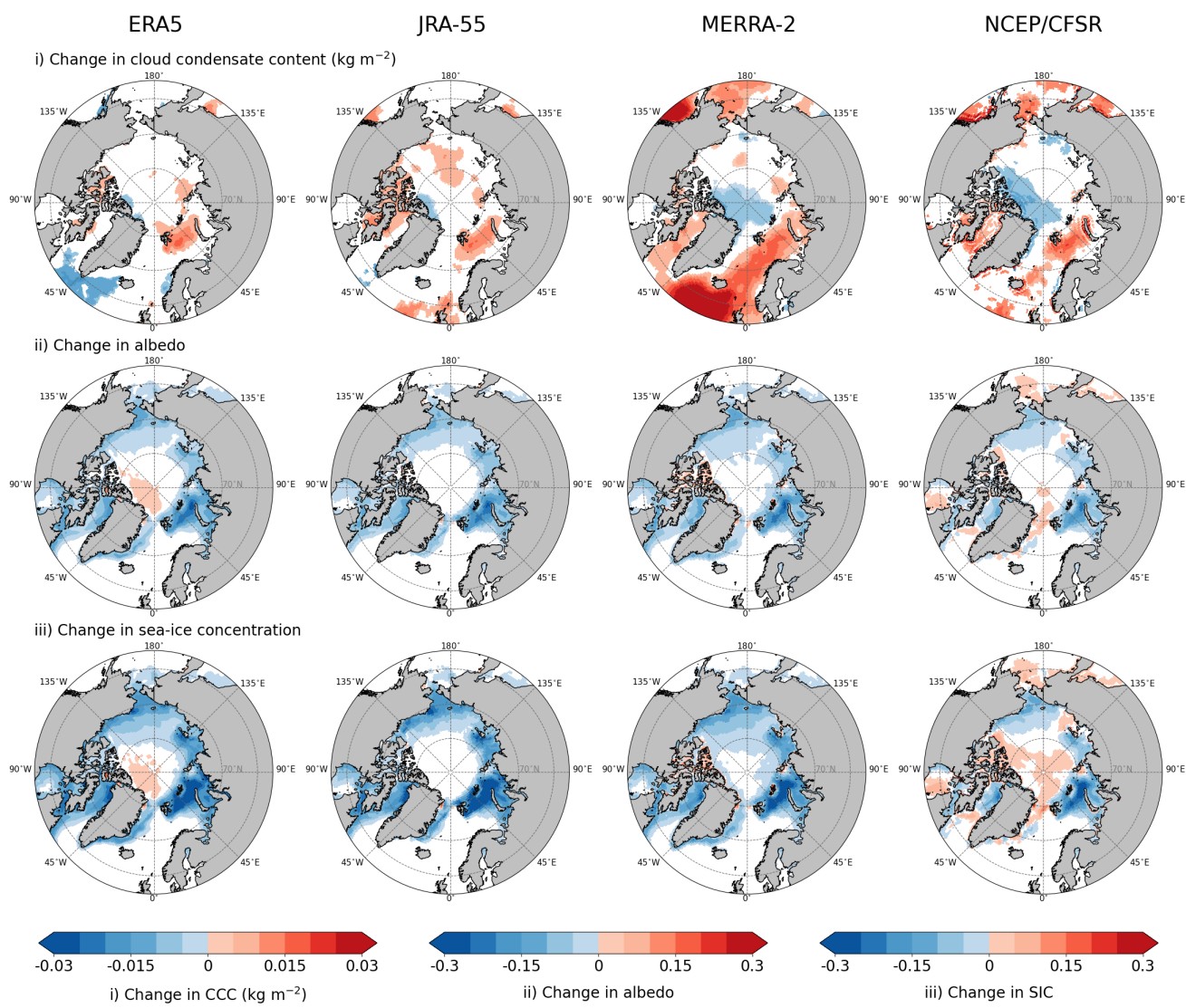

**Figure 6.** Changes in decadal means (calculated from daily means) 2001–2021 minus 1980–2000, May–June–July. Row **(i)** shows cloud condensate content (CCC, vertically integrated cloud liquid water + ice), row **(ii)** shows surface albedo, and row **(iii)** sea-ice concentration (SIC). Only statistically significant results at the 5 % level of significance are shown (insignificant masked in white). In rows **(ii)** and **(iii)**, values within an interval (-0.01,0.01) are also masked in white.



## 4  Discussion

### 4.1  Differences between reanalyses in the effects of sea-ice concentration on surface upward longwave radiation

Our results indicated generally the highest sensitivity of ULW to SIC in the cold season November–April in the Central Arctic (Figs. 1 and S1) — over 150 W m$^{-2}$ ULW per -0.1 change in SIC. Utilizing JRA-55 data, the linear bilateral ODR model did not converge in large areas due to the data set's binary representation of the sea ice, however, in the areas of adjacent Arctic seas, where the model returned values of the slope between the two variables, the results were in agreement with other reanalyses. In the warm season May–October, the sensitivity of ULW to SIC was generally lower, up to 80 W m$^{-2}$ ULW per -0.1 change in SIC (Figs. S3 and S4). The magnitude of the highest sensitivities of ULW to SIC was similar amongst reanalyses, though their spatial extent somewhat differed.

The effect of Arctic SIC on ULW mostly decreased in all seasons between 1980–2000 and 2001–2021 due to SIC decline and warming of the sea-ice surface, however, we noted also an increased sensitivity of ULW to SIC in the Central Arctic, north and northeast of Greenland. As shown in Fig. 2 (Point 2) the daily SIC in November–December–January increased in the Atlantic sector of the Central Arctic between 1980–2000 and 2001–2021 in ERA5, MERRA-2, and NCEP/CFSR. Fig. S10 row iii, also indicates statistically significant decadal increase in SIC in this area in February–March–April. Greater daily SIC is then directly connected to increased sensitivity of ULW to SIC. The SIC increase may be related to thinning of the Arctic sea ice, which reduces the internal resistance of the ice field, allowing certain atmospheric and oceanic forcing to generate faster ice drift (Leppäranta, 2011). Higher drift speeds along the Transpolar Drift Stream (TDS) favour increased accumulation of ice north of Greenland (Kwok, 2015), resulting in increased SIC. Another potential factor favouring faster ice drift is increased wind speeds along TDS (Smedsrud et al., 2017). However, trends in the wind speeds are sensitive to the region and period addressed (Spreen et al. (2011);Vihma et al. (2012)).

Comparing the variance in ULW explained by SIC and $T_{ice}$ in November–April in ERA5, MERRA-2, and NCEP/CFSR, we found very good agreement between these three data sets in that variance in $T_{ice}$ was mostly responsible for variance in ULW ($R^2$ over 90 %) in areas, where SIC was very high. In the marginal ice zone, however, SIC explained around 60 % of variance in ULW while $T_{ice}$ only up to 30 % (Figs. 3, S5).

### 4.2  Differences between reanalyses in the effects of sea-ice concentration and clouds on surface upward shortwave radiation

We found positive correlation between SIC and USW in all seasons and both study periods as expected. Similarly to the effect of SIC on ULW, the effect of SIC on USW also mostly weakened between 1980–2000 and 2001–2021. While the sea-ice and its surface albedo decline plays undeniable role in the weakening of this effect, also decadal changes in DSW must be considered when assessing decadal changes in USW.

Considering May–June–July, we found the magnitude of decadal change in mean USW (more than -25 W m$^{-2}$) and its spatial pattern similar among reanalyses in adjacent Arctic seas, however, this variable somewhat differed in the Central Arctic (Fig. 5e–h). Namely ERA5 and NCEP/CFSR showed decadal increase in the mean USW (around +10 W m$^{-2}$) north of Green-



land and Canadian archipelago, whereas JRA-55 indicated decadal reduction in the mean USW (around -10 W m$^{-2}$) where
other reanalyses did not show significant changes. As the magnitude of changes in USW explained by changes in DSW ($\Delta$
USW$_{DSW}$) was very similar among all reanalyses, the differences between these data sets in decadal seasonal changes in USW
were mostly attributable to decadal changes in the surface albedo (Figs. 5, S9, S11). These results are similar to those of Cao
et al. (2016) who considered the surface albedo product from the Satellite Application Facility on Climate Monitoring clouds,
albedo, and radiation data set (CLARA-SAL) additionally to reanalyses data from 1982–2009. According to their findings,
JRA-55 data agreed the best with the satellite observations, which did not show any increase in annual surface albedo in the
Central Arctic, north of Greenland and Canadian archipelago that we saw in ERA5 and NCEP/CFSR data.

In our analyses of decadal seasonal changes, we found the largest differences between reanalyses in CCC (row i in Figs. 6,
S10, S12). However, in all seasons, the magnitude of $\Delta$ USW$_{DSW}$ was very similar among all reanalyses, so from the point of
view of solar radiation, clouds did not seem to be a key factor for the inter-reanalysis differences in decadal seasonal changes.

## 4.3 The role of clouds on surface radiative fluxes and their differences between reanalyses

The clouds in the Arctic have typically positive net radiative effect on the surface for most of the year, as they have more
impact by emitting longwave radiation towards the surface (DLW) and warming it than cooling it by reflecting the shortwave
radiation back to space (Wendish et al. (2019); Morrison et al. (2019)). In May–June–July, however, incoming solar radiation
in the Arctic is very high and clouds regulate the melting of sea ice and partly offset the strength of the sea ice–albedo feedback
(Choi et al., 2020). The sign and strength of the radiative effect of clouds mostly depend on the cloud fraction, longevity,
opacity (liquid/ice phase partitioning), and temperature of the cloud layer. The presence and properties of clouds have potential
to considerably affect the surface and near-surface temperature and humidity. As we showed in Figs. 3 and S5, in areas with
high SIC, changes in T$_{ice}$ are important for explaining the variance in ULW in November–April, and these may be to a large
part driven by changes in clouds. At the same time, SIC also affects the formation of clouds, via turbulent surface fluxes of
sensible and latent heat. As shown in observational studies by Palm et al. (2010) and Liu et al. (2012) and in the study of
Schweiger et al. (2008) who used reanalysis data from ERA40 (predecessor of ERA5), cloud cover variability near the sea ice
margins is strongly linked to sea-ice variability and areas with increased mid-level cloudiness coincide with those of recent
sea-ice decline. Also in our results, throughout the seasons, we saw the decadal increase in CCC in areas of strong SIC decline,
although, reanalyses did not always agree on the magnitude or spatial extent of this increase. The increase in CCC is in line
with Sledd and L'Ecuyer (2021).

Despite their importance for the Arctic surface energy budget, the clouds appear to be one of the largest sources of uncer-
tainty as a variable in reanalyses and as a component of the Arctic climate system. This is mostly because the retrieval of cloud
fraction and cloud properties (such as optical depth, top pressure, or cloud condensate content) from satellite measurements
includes considerable uncertainties when using different sensors or even different approaches to derive the data from measured
radiances (Devasthale et al., 2020). Also the insufficiency of supporting ground-based observational network in the Arctic
contributes to the uncertainties. In our study, we only calculated decadal seasonal differences in mean CCC, however, even by
using this simple calculation and just one cloud parameter, we noted the spread in values between the reanalyses (row i in Figs.



6, S10, S12).

In reality, aerosols affect the radiative properties of Arctic clouds (Garrett and Zhao, 2006). These effects have under-gone notable changes due to shifts in aerosol sources and regional atmospheric conditions (Warneke et al. (2010); Stohl et al. (2013)). Among the reanalyses applied in this study, MERRA-2 is based on daily assimilation of aerosol data, whereas ERA5, JRA-55, and NCEP/CFSR apply climatological aerosol concentrations. In principle, it should be possible to distinguish the contribution of aerosols to the radiative transfer and its seasonal and decadal changes, however, the output available from the

reanalyses is not sufficient for such analyses.

## 4.4 The role of surface albedo and its differences between reanalyses

Surface albedo (the proportion of incident shortwave radiation that is reflected back to space by the surface) is a key component of Arctic climate system. This property of the surface is the most important in May–June–July when the incoming shortwave radiation peaks and low albedo allows a much larger part of it to penetrate into (and warm) the surface. While the snow and sea

ice and their properties control the surface albedo, at the same time, surface albedo controls the mass balance of snow and sea ice. This effect has a seasonal cycle, when (1) the bare sea ice with large amount of melt ponds and lower albedo during the melt season accelerates further ice melt by allowing more shortwave radiation to be absorbed, while (2) the dry snow on top of the sea ice generates greater surface albedo before and after the melt season, protecting the sea ice from shortwave radiative warming. Pistone et al. (2014) showed the close relationship of SIC and surface albedo in satellite data from The Clouds and

Earth's Radiant Energy System (CERES) and the Special Sensor Microwave Imager (SSM/I) and our results demonstrated that the patterns of diminishing SIC coincided with the patterns of the surface albedo decrease (rows i and iii in Figs. 6, S10, S12).

    The albedo of the sea ice is parameterised in JRA-55 and NCEP/CFSR, considering summer melt ponds and surface temperature, whereas in ERA5 and MERRA-2, it has a prescribed seasonal cycle that is the same for the whole study period of our analyses. Pistone et al. (2014) observed pan-Arctic darkening with clear-sky albedo decreasing from 0.39 to 0.33 and all-

sky albedo decreasing from 0.54 to 0.48 during 1979–2011. These findings and their consequences for the prescribed surface albedo in reanalyses are demonstrated in the the comparison study by Pohl et al. (2020), who utilized satellite data from Medium Resolution Imaging Spectrometer (MERIS) to derive the albedo of Arctic sea ice. In their analyses, utilizing data from May to September 2003–2011, ERA5 was found to generally overestimate the albedo of first-year ice and underestimate the albedo of multiyear ice. Overestimation of the albedo likely happens due to not accounting (1) for the warming of the sea ice and (2) for

the increasing amount of melt ponds on top of the sea ice during the melt seasons in recent decades. Underestimation of the albedo may be connected to sea-ice thickness prescribed in ERA5 (1.5 m) that, in reality, is likely higher in the multiyear ice in the Central Arctic, north of Greenland generating higher real albedo measured by MERIS.



## 5 Conclusions

In the present study, we quantified the uncertainties in the effects of Arctic sea-ice concentration on surface radiative fluxes
as represented in four atmospheric reanalyses, a complement to Uhlíková et al. (2024), where we addressed turbulent surface
fluxes of sensible and latent heat. Our results showed the greatest sensitivity of surface upward longwave radiation to SIC in
the cold season November–April (over 150 W m$^{-2}$ per -0.1 change in SIC) and greatest sensitivity of surface upward shortwave
radiation to SIC in May–July (over 100 W m$^{-2}$ USW per 0.1 change in SIC). We found that the effect of SIC on both surface
upward longwave and shortwave radiation has mostly weakened in all seasons between the study periods of 1980–2000 and
2001–2021. Unlike in the case of the effects of SIC on turbulent surface fluxes, we did not find particularly higher sensitivity
of surface upward radiative fluxes to SIC in NCEP/CFSR (which includes both modelled sea-ice thickness and snow depth on
the sea ice and accounts for their insulating effects) compared to other reanalyses (which assume a constant sea-ice thickness
and do not account for the snow on sea ice).

Furthermore, we analysed decadal changes in surface downward and upward shortwave radiation and quantified differences
among reanalyses in these variables and additionally in the surface albedo, sea-ice concentration, and cloud condensate content.
These analyses indicated that approximately 80 % of the decadal decrease in upward shortwave radiation in May–July was
caused by a decrease in surface albedo, controlled by SIC decrease, and the rest was caused by a decrease in downward
shortwave radiation due to increase in cloudiness, mostly close to sea ice margins. CCC showed the largest uncertainty among
reanalyses in all seasons, however, the magnitude of decadal changes in surface upward shortwave radiation explained by
changes in surface downward shortwave radiation was very similar among reanalyses. Accordingly, from the point of view of
solar radiation, clouds did not seem to be a key factor for the inter-reanalysis differences in decadal seasonal changes.

Expanding quantitative knowledge on differences in the representation of the Arctic surface energy budget in atmospheric
reanalyses is needed, because the Arctic amplification of climate warming is primarily surface-based (Serreze et al. (2009);
Taylor et al. (2022)) and reanalyses are broadly utilized and relied upon in studies on past climate and related processes in the
Arctic.



*Code and data availability.*  https://doi.org/10.5281/zenodo.11565044 (Uotila, Uhlíková, 2024), and https://a3s.fi/uhlitere-2000789-pub/* (last access: 11 June 2024) (Hersbach et al., 2023; Japan Meteorological Agency, 2013; GMAO, 2015a, b, c; Saha et al., 2010b, 2011). (To download a desired file, the name of it must be entered after the last forward slash, instead of *. Names of files can be found in codes. Data description can be found at https://a3s.fi/uhlitere-2000789-pub/README2_data2.odt, last access: 13 May 2024.

*Author contributions.*  TU prepared the manuscript with contributions of TV, PU, and AYK. TV, PU, and AYK designed the concept of the study with contributions of TU. PU developed the code with the contribution of TU. TU collected and processed data and performed analyses.

*Competing interests.*  The authors declare that they have no conflict of interest.

*Disclaimer.*  Neither the European Commission nor ECMWF is responsible for any use that may be made of the Copernicus information or data it contains.

*Acknowledgements.*  During most of the work, TU was a university-funded doctoral researcher at the University of Helsinki. The work of AYK, PU, TV and, in the final stage, TU was supported by the European Commission's Horizon 2020 Framework Programme (PolarRES; grant no. 101003590).
Hersbach et al. (2023) was downloaded from the Copernicus Climate Change Service (C3S) Climate Data Store (2024). The results contain modified Copernicus Climate Change Service information 2023. Furthermore, we acknowledge the providers of the data of the other three
reanalyses used in our study: Japan Meteorological Agency, the National Center for Atmospheric Research (JRA-55, NCEP/CFSR, CFSv2), and the Global Modeling and Assimilation Office (MERRA-2).



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
