# Peer review of "Effects of Arctic sea-ice concentration on surface radiative fluxes in four atmospheric reanalyses"

_EGUsphere, 2024_

## Author Comment (AC1)

Dear Reviewer,

thank you very much for your time and suggestions for improving our manuscript. Please see our responses (*in italics*) to your comments (**in bold**) below:

**Major Comment:**

**1) The variance of upward LW radiation to sea ice is greatest at the 100% sic level (Fig. 2). This, however, is to be expected because this is a limiting value for sea ice, as long as it is cold enough for sea ice to remain at 100% the conditions can vary greatly, so this value should not be included in a sensitivity analysis as it no longer reflects a sensitivity to sea ice. If included in the results, it will steepen the slope artificially. I recommend the authors redo the analysis including only sic values from 1-99% and test whether this changes any of the results presented in the manuscript.**

*We agree that this would be the case, if SIC was only controlled by air temperature. However, SIC also depends on the wind, which can result in ice drift convergence (increase of SIC) or divergence (decrease of SIC). Hence, the ice surface temperature, controlling upward LW radiation, does not necessarily differ much between conditions of 1 and almost 1 SIC.*

*In Figure 2, Point 2, there are 4-5 hexagonal bins between SIC 0.95-1 and the range of ULW seems to be rather similar at least among the rightmost 2-3 values of SIC (~0.98-1).*

*From Figure 2, Point 1, we used MERRA-2 data from NDJ 2001-2021 as an example, and compared the slopes of the regression lines in the original case (all values of SIC) and only days with SIC 0.1-0.99. In the Figure below, we show, that the slopes of the regression lines differ only very little between the two cases.*

[Figure]

*We note that for the comparison depicted above, we utilized ordinary-least-square regression analysis to obtain the slope and intercept of the regression line, because the ODR model did not converge in Point 1 when using SIC 0.01-0.99 data.*

*Because of the different method used, the slope of the regression line in original data (left panel in the figure above) is somewhat different (less steep) than in our original manuscript (fifth panel in Figure 2, Point 1). However, as we used the same method for both cases in the figure above, it does not affect the results of the comparison.*

**2) Furthermore, the sensitivity results (e.g. Fig. 2) reflect a sensitivity that is larger at high values of sic than low values. This makes sense as low values of sic tend to have thinner sea ice and a greater percentage of open ocean. For smaller values of sic, the open ocean serves as a moderating influence that lowers the variance of the sea ice temperature and upward LW flux due to its high thermal inertia. For thinner sea ice, the ocean below also moderates the variance. The greater the sea ice thickness (SIT) the weaker the influence the ocean below has on the surface temperature and upward LW flux. Therefore, I suggest the authors attempt to include an analysis of SIT for the sensitivity analysis, which admittedly might be cumbersome as SIT observations are still lacking but a reanalysis such as PIOMAS might be helpful. This would provide a more insightful analysis of the sensitivity of upward LW fluxes to sea ice, while providing for a more complete physical explanation of the results.**

*We added a new figure to subsection 3.1 of the revised manuscript to address this comment. We used February data from the SHEBA campaign and carried out calculations on the effects of sea ice thickness and snow depth on the conductive heat flux from the ice base to snow surface, which further affects the surface temperature and upward LW flux. The results are presented in the new figure and interpreted in the revised text.*

**Additional Comments:**

**3) The results section 3, discussion section 4 and conclusions section 5 have some repetitive information. I suggest a revision to streamline the paper.**

*We reviewed sections 3, 4, and 5 for repetitive information and reduced their amount whenever possible.*

**4) Lines 88-89: Sentence is unclear. Improve the clarity of the definitions for R1 and R2, respectively.**

We rephrased the part of the text as: '*...T stands for number of days in one sample (in our case days in seasons in the periods of 1980-2000 or 2001-2021), $R_1$ for correlation coefficient of lag 1 auto-correlation of SIC, and $R_2$ for correlation coefficient of lag 1 auto-correlation of surface radiative flux (ULW or USW).*'

**5) Lines 100-101: Why is the open ocean 'usually' and not always warmer than the sea ice surface? If open ocean is colder than sea ice, wouldn't we expect the ocean to freeze into sea ice?**

*Open ocean in the Arctic can be colder than the sea ice during the melting season by the following mechanism: Water under the sea ice is usually close to the sea-water freezing point -1.8 °C (unless e.g. warm upwelling of ocean water present). Then, when a lead opens due to divergent ice motion, this temperature becomes the ocean surface temperature, while the sea-ice surface next to the lead may be close to the melting temperature of snow and ice (0 °C).*

**6) Lines 135-136: Did the authors mean "cannot be applied where no SIC is present"? I suggest a revision.**

We rephrased the part of the text as: '*This assumption cannot be applied in the warm season (May-October) in the majority of adjacent seas outside the Central Arctic, because the surface temperature of the ocean is likely often higher than -1.8 °C, therefore we focused on the cold season (November-April) in these analyses. We are also aware, that in the Greenland and Barents seas, even cold-season ocean temperature may be warmer than -1.8 °C due to the North Atlantic Current carrying warm Atlantic water.*'

**7) Line 206: Suggest change to "we also noted".**

*We changed the text accordingly.*

**8) Line 224: Suggest change to "plays an undeniable role" and from "also decadal …" to "decadal changes in DSW must *also* be…".**

*We changed the text accordingly.*

---

## Author Comment (AC2)

Dear Reviewer,
thank you very much for your time and suggestions for improving our manuscript. Please see our responses (*in italics*) to your comments (**in bold**) below:

**This study aimed to compare the affects of sea ice concentration on the surface radiative fluxes from 4 reanalysis and this sort of a companion paper to their previous study looking at turbulent fluxes from these reanalysis. They found that the upward longwave radiation was most sensitive to SIC in the winter, and upward shortwave radiation was most sensitive to SIC during the summer. They found that the relationship between SIC and upward longwave radiation has decreased from the first 20 years compared to the last 20 years in the 1980-2021 record and attributed this to the thinning ice and warming surface temperatures.**

**I found this study to be very insightful, and the methods and results were clearly explained and easy to follow. The figures were also clear and easy to understand. I feel like these types of analysis are essential to the scientific community to better understand the uncertainties and limitations posed by these global reanalysis products when studying the Polar regions like the Arctic Ocean. I feel like this paper should be published after my minor comments are addressed.**

**9) Sentence beginning on line 30: Needs a citation**

*We added a citation to Persson (2012).*

**10) Line 70: why do you not also include surface downward longwave radiation? I am sure you would want to do this to account for the full radiative fluxes.**

*Downward longwave (as well as shortwave) radiation depends on cloud formation and properties, and emisivity of the atmosphere more than on SIC (which is in the center of attention of our manuscript). SIC can and does contribute to some degree to a cloud formation by a flux of moisture to the atmosphere in areas of recent sea-ice decline, and vice versa to lower cloud formation in areas with increased SIC (indicated in our Figure 6 and Lines 250–253 in Discussion of the original manuscript). However, as shown e.g. in Nygård et al., 2020 (their Figure 6), the moisture from evaporation in the areas with sea ice present contributes only very little to cloud water and total column water vapour compared to horizontal moisture transport, which dominates the regional moistening pattern in the Arctic.*

*For these reasons and as the manuscript is already rather long, we decided to exclude analyses regarding DLW radiation and include the DSW radiation only as a part of the explanation of changes in USW radiation between our two study periods.*

**11) Table 1: can you elaborate more on how the albedo is parameterized for NCEP/CFSR**
**Is it coupled to the ocean or atmosphere or both? Please elaborate. Any information on the sea ice model that they use? It might be more specific if you actually spelled out**

**what the albedos were for the other reanalysis rather than just stating the citations, and include this information in the table.**

*The albedo in CFSR and CFSv2 comes from Sea Ice Simulator 1 (SIS-1) by Geophysical Fluid Dynamics Laboratory (GFDL). Sea ice plays the role of a general interface between the atmosphere and the ocean in this model.*

*CFSR and CFSv2 use both atmospheric and ocean model (GFS and MOM4).*

*We added this information to Table 1.*

*To concisely describe the seasonal cycle or parameterization of albedo in a table is a rather difficult task. Hence, we decided to use an example and calculate the average albedo for the grid cell nearest to the North Pole for month of June in the years 1990 and 2010 to offer a comparison of this variable between the reanalyses. We present the results in the revised manuscript.*

**12) Figure 3, I think that makes sense because we can probably assume that the ocean surface temperature in the marginal ice zones is likely above -1.8C and the ice is less compact so the ice temperature that was calculated was probably off some.**

*This comment seems unfinished, but if it was meant as a possitive comment, we thank the reviewer.*

**13) Line 194: Yes this makes sense that the CCC would be so different between all of the reanalyses because they all have differing cloud schemes (one moment, two moment, etc). It might be nice to reference these differences or add their parameterizations to the table 1.**

*Clouds in our study are only touched rather briefly for the purpose of explaining part of decadal changes in upward SW radiation by decadal changes in downward SW radiation, which depend on cloud formation and properties.*

*Cloud parameterization in reanalyses is a very komplex subject and as mentioned in the response to comment 10, our main focus for this manuscript ended up being mainly surface upward radiative fluxes, therefore we do not believe that going further into cloud parameterizations fits the predominant topic of the study.*

*We do consider the subject of cloud parameterization in reanalyses crucial for understanding their differences in the representation of the Arctic climate system (as indicated in the subsection 4.3 of Discussion in the original manuscript).*

**14) Figure 6: Any idea why MERRA2 has such a large change in CCC, especially in the North Atlantic, compared to other reanalysis?**

*One reason for larger cloud condensate content between 1980-2000 and 2001-2021 in MERRA-2 in the North Atlantic south and southwest of Iceland (which we found to some extent also in February-March-April and August-September-October; Figures S10, S12) may be that MERRA-2 assimilates aerosol observations, while the other reanalyses only apply climatological aerosol concentrations.*

*Other reasons are probably related to different parameterization of cloud microphysics between MERRA-2 and other reanalyses.*

*As our study area of focus was limited to the ice-covered seas, we generally did not assess areas more to the south even though they are depicted in some of our figures.*

**General Comment:**

**15) Since the SIC is so important for the energy budget of the lower atmosphere and ocean in the Arctic, it might be good to compare the SIC with passive microwave SIC observations to determine which SIC is most realistic? Then what conclusions can be made towards your other results which are so highly SIC dependent/driven?**

*Comparison of reanalyses to observations is undoubtedly necessary in order to assess their accuracy. However, SIC in all four reanalyses in our study is based on information from satellite passive microwave sensors.*

*There is no consensus on which passive-microwave-based SIC data set is most realistic as different passive microwave sensors are utilized and even for thr same set of raw data, SIC depends on the processing algorithm applied (such as NASA-Team, Bootstrap, and ARTIST algorithms). Hence, we cannot answer the second question.*

*In principle, the suggested comparison could be based on Synthetic Aperture Radar (SAR) data, which has a very high resolution and identifies each pixel as either sea ice or open water, allowing calculation of regional sea-ice concentrations. However, such comparison would require a major work (of the order of several months) and also SAR data have challenges in the pixel identification.*

References:

Nygård, T., Naaka, T., and Vihma, T.: Horizontal Moisture Transport Dominates the Regional Moistening Patterns in the Arctic, J. Climate, 33, 6793–6807, https://doi.org/10.1175/JCLI-D-19-0891.1, 2020.

Persson, P. O. G.: Onset and end of the summer melt season over sea ice: Thermal structure and surface energy perspective from SHEBA, Clim. Dyn., 39, 1349–1371, 2012.

---

## Author Response (AR2)

Dear Reviewer,
thank you very much for your time and suggestions for improving our manuscript. Please see our responses (*in italics*) to your comments (**in bold**) below:

**1) Line 72: What is meant by "All surface radiative fluxes were defined as positive."? There are downward and upward surfaces flux. Does this mean both downward and upward surface radiative fluxes are defined as positive? If yes, I suggest making this explicit, as the norm is to define either upward or downward as positive and the other as negative.**

*Yes, it is meant that both downward and upward radiative surface fluxes are positive (they also come in this shape from all the reanalyses). We adjusted the text to: 'All surface radiative fluxes (both upward and downward) were defined as positive.'* Line 72

**2) Line 149: I suggest deleting 'also'.**

*We adjusted the text accordingly.* Line 150

**3) Lines 217-218: I suggest changing "also reduction of DSW…" to "the reduction of DSW also played a role."**

*We adjusted the text accordingly.* Line 218–219

**4) In Section 4.1 the authors discuss how the increase in SIC in the central Arctic leads to increased sensitivity in ULW. However, why does the increase in SIC in the central Arctic not lead to strengthening of the sensitivity in USW? This should be discussed in section 4.2 to contrast with section 4.1.**

*We added a specification to Section 4.1 that we are addressing the increase in decadal sensitivity of ULW to SIC due to decadal increase of SIC in November–December–January (that we saw in Fig. 1).*

*If we compare the decadal change in ULW sensitivity to SIC in May–June–July (Fig. S3), it shows very similar patterns to the decadal change in USW sensitivity to SIC in May–June–July (Fig. 5).*

*Additionally, to show the decadal change of the daily SIC and USW in the Central Arctic/Greenland Sea in this season, we present representative grid cells from ERA5 and MERRA-2 (similar to Point 2 in Fig. 2) in Fig. X below. In both reanalyses and both study periods, the daily SIC was mostly around 0.9, however, we noted an increase in SIC below 0.8 in the second study period leading to weaker effect of SIC on USW in May–June–July in 2001–2021 compared to 1980–2000. However, we do not consider the above results interesting enough to be discussed in Section 4.2. We understand the Reviewer's point of view. Had the increased sensitivity of ULW to SIC and the decreased sensitivity of USW to SIC occurred in the same season, it would require discussion in Section 4.2. Now when it is clarified that the increased sensitivity of ULW to SIC occurred in November–December–January (when there is very little to none solar radiation), the discussion is not relevant.*

[Figure]

[Figure]

**Figure X.** Daily sea-ice concentration (SIC) and upward shortwave radiative flux (USW) in selected grid cell (nearest to 81◦ N, 0◦ W). ERA5 and MERRA-2 data, days in May–June–July (1932 days). Black solid lines depict (a part of) the regression lines and illustrate their slope.

**5) Lines 298-299: Why doesn't the large spread in clouds have an impactful effect on the solar radiation? Hypothesis or additional insight should be added.**

Text in question:
'In our study, we only calculated decadal seasonal differences in mean CCC, but even by using this simple calculation and just one cloud parameter, we noted a large spread in values between the reanalyses (row i in Figs. 7, S11, S13). However, in all seasons, the magnitude of changes in USW explained by changes in DSW ($\Delta USW_{DSW}$) was very similar among reanalyses (panels i–l in Figs. 6, S10, S12), so from the point of view of solar radiation, clouds did not seem to be a key factor for the inter-reanalysis differences in decadal seasonal changes.'

*Thank you for pointing out this part of the text. We noted the confusing phrasing.*

*Large spread among reanalyses in decadal changes in cloud condensate content (CCC, for May–June–July shown in Fig. 7 row i) is reflected in the spread in decadal changes in downward solar radiation at the surface (DSW, Fig. 6a–d). It seems that we meant to point out that the effect of CCC on USW (via $\Delta USW_{DSW}$, Fig. 6i–l), however, does not show a very large spread. After reconsidering, we do not see any special reason why this should be mentioned and decided to not include the text on Lines 297–299 in the revised manuscript.*
*The key message regarding the effect of CCC (via $\Delta USW_{DSW}$) on USW should be that, according to our results, it is smaller than the effect of decadal changes in surface albedo ($\Delta USW_b$) on USW as shown in Fig. 6i–l and m–p. Such message was already mentioned in the Section 3.3 of the Results and we do not consider it necessary to mention it again in the Discussion.*